# Artificial Sweeteners: A Double-Edged Sword for Gut Microbiome

**DOI:** 10.3390/diseases13040115

**Published:** 2025-04-15

**Authors:** Helal F. Hetta, Nizar Sirag, Hassabelrasoul Elfadil, Ayman Salama, Sara F. Aljadrawi, Amani J. Alfaifi, Asma N. Alwabisi, Bothinah M. AbuAlhasan, Layan S. Alanazi, Yara A. Aljohani, Yasmin N. Ramadan, Noura H. Abd Ellah, Abdelazeem M. Algammal

**Affiliations:** 1Division of Microbiology, Immunology and Biotechnology, Department of Natural Products and Alternative Medicine, Faculty of Pharmacy, University of Tabuk, Tabuk 71491, Saudi Arabia; habdelgadir@ut.edu.sa; 2Division of Pharmacognosy, Department of Natural Products and Alternative Medicine, Faculty of Pharmacy, University of Tabuk, Tabuk 71491, Saudi Arabia; nmona@ut.edu.sa; 3Department of Pharmaceutics, Faculty of Pharmacy, University of Tabuk, Tabuk 71491, Saudi Arabia; agrawan@ut.edu.sa; 4PharmD Program, Faculty of Pharmacy, University of Tabuk, Tabuk 71491, Saudi Arabia; 421000534@stu.ut.edu.sa (S.F.A.); 421001486@stu.ut.edu.sa (A.J.A.); 421000741@stu.ut.edu.sa (A.N.A.); 421003257@stu.ut.edu.sa (B.M.A.); 421002588@stu.ut.edu.sa (L.S.A.); 421001273@stu.ut.edu.sa (Y.A.A.); 5Department of Microbiology and Immunology, Faculty of Pharmacy, Assiut University, Assiut 71515, Egypt; yasmine_mohamed@pharm.aun.edu.eg; 6Department of Pharmaceutics and Pharmaceutical Technology, Faculty of Pharmacy, Badr University in Assiut, Naser City 2014101, Assiut, Egypt; nora.1512@aun.edu.eg; 7Department of Pharmaceutics, Faculty of Pharmacy, Assiut University, Assiut 71515, Egypt; 8Department of Bacteriology, Immunology and Mycology, Faculty of Veterinary Medicine, Suez Canal University, Ismailia 41522, Egypt

**Keywords:** gut microbiome, artificial sweeteners, eubiosis, dysbiosis

## Abstract

**Background and Aim**: The human gut microbiome plays a crucial role in maintaining health. Artificial sweeteners, also known as non-nutritive sweeteners (NNS), have garnered attention for their potential to disrupt the balance of the gut microbiome. This review explores the complex relationship between NNS and the gut microbiome, highlighting their potential benefits and risks. By synthesizing current evidence, we aim to provide a balanced perspective on the role of AS in dietary practices and health outcomes, emphasizing the need for targeted research to guide their safe and effective use. **Methods**: A comprehensive literature review was conducted through searches in PubMed and Google Scholar, focusing on the effects of artificial sweeteners on gut microbiota. The search utilized key terms including “Gut Microbiome”, “gut microbiota”, “Eubiosis”, “Dysbiosis”, “Artificial Sweeteners”, and “Nonnutritive Sweeteners”. **Results**: NNS may alter the gut microbiome, but findings remain inconsistent. Animal studies often report a decrease in beneficial bacteria like *Bifidobacterium* and *Lactobacillus*, and an increase in harmful strains such as *Clostridium difficile* and *E. coli*, potentially leading to inflammation and gut imbalance. Disruptions in short-chain fatty acid (SCFA) production and gut hormone signaling have also been observed. However, human studies generally show milder or no significant changes, highlighting the limitations in translating animal model findings directly to humans. Differences in study design, dosage, exposure time, and sweetener type likely contribute to these varied outcomes. **Conclusions**: While NNS offer certain benefits, including reduced caloric intake and improved blood sugar regulation, their impact on gut microbiome health raises important concerns. The observed reduction in beneficial bacteria and the rise in pathogenic strains underscore the need for caution in NNS consumption. Furthermore, the disruption of SCFA production and metabolic pathways illustrates the intricate relationship between diet and gut health.

## 1. Introduction

Artificial sweeteners (AS) have become integral to modern diets, driven by increasing health awareness and the global rise in obesity, diabetes, and metabolic syndrome [1]. These low- or non-caloric substitutes for sugar promise sweetness without the caloric burden, offering a potential solution to reduce sugar intake while addressing health challenges. Widely incorporated into beverages, processed foods, and even medications, AS have reshaped the nutritional landscape. However, their physiological impact goes beyond their calorie-free profile, raising critical questions about their broader effects on human health, particularly through interactions with the gut microbiome. Although usually considered safe, some people have expressed concerns about the long-term safety of particular sweeteners [2,3,4,5,6].

The gut microbiome, a complex ecosystem of trillions of microorganisms, plays a pivotal role in maintaining metabolic and immune homeostasis. Alterations in its composition or functionality are referred to as dysbiosis. Dysbiosis entails changes in composition and diversity such as: (1) overgrowth of conditionally pathogenic microbes and metabolites, (2) decrease in conditionally beneficial microbes and metabolites, (3) alterations in host intestinal function due to aberrant colonization or inflammation. This disruption has been linked to a range of adverse health outcomes, including inflammation, metabolic disturbances, and increased susceptibility to diseases such as inflammatory bowel disease (IBD) and type 2 diabetes [7,8,9,10,11]. While AS were initially deemed biologically inert due to their limited absorption in the gastrointestinal tract, emerging studies challenge this notion. Evidence suggests that even negligible amounts of AS reaching the gut can influence microbial diversity and activity, with potential downstream effects on health [12].

Recent research has revealed paradoxical roles for AS. On one hand, they serve as an effective tool in dietary management by supporting glycemic control and calorie reduction [13]. On the other hand, their consumption has been linked to gut dysbiosis, which can disrupt metabolic signaling pathways and immune responses [14,15]. Studies demonstrate that AS may reduce beneficial bacterial populations, such as Lactobacillus and Bifidobacterium, while promoting pathogenic strains, potentially exacerbating gut inflammation and increasing susceptibility to disease [16,17]. These findings underscore the need for a nuanced understanding of the dual effects of AS on gut health.

This review explores the complex relationship between AS and the gut microbiome, highlighting their potential benefits and risks. By synthesizing current evidence, we aim to provide a balanced perspective on the role of AS in dietary practices and health outcomes, emphasizing the need for targeted research to guide their safe and effective use.

## 2. Current Status of Knowledge

### 2.1. Gut Microbiome Community

The balance of the intestinal microbiome is directly related to human health and disease. The human GI tract has a dense microbial ecosystem that consists of bacteria, viruses, archaea, eukarya, and parasites [8,18], with approximately 100 trillion microbes [19]. Extensive research has been conducted to highlight the critical interaction between gut microbiome and fundamental human biological processes. For instance, recent research has demonstrated that the human microbiome plays a critical role in immunity, metabolism, and nutrient extraction [20]. The composition of the microbiota differs from person to person and can be classified as either dysbiosis, a microbial imbalance that might result in pathological consequences, or eubiosis, a healthy microbiota with a balance of beneficial bacterial species [21]. Each host has a distinct microbial community, which can fluctuate quickly depending on several environmental conditions, including nutrition [12].

The gut environment encourages the proliferation of bacteria from seven major divisions: *Firmicutes*, *Bacteroidetes*, *Actinobacteria*, *Fusobacteria*, *Proteobacteria*, *Cyanobacteria*, and *Verrucomicrobia* [22,23] (Figure 1). *Bacteroidetes* and *Firmicutes* are the predominant bacterial communities in the gastrointestinal system, accounting for more than 90% of the total inhabitants [24]. The phylum *Firmicutes* contains more than 200 genera of Gram-positive bacteria, including *Clostridium*, *Eubacterium*, and *Ruminococcus* [25]. *Staphylococci*, *Enterococci,* and *Lactobacilli* are among the less abundant genera in this phylum [24]. The phylum *Bacteroidetes* contains about 7000 distinct species of Gram-negative bacteria, principally from the genera *Bacteroides*, *Parabacteroides*, *Prevotella,* and *Alistipes* [25]. The phylum *Actinobacteria* is less prevalent than *Bacteroidetes* and *Firmicutes* and primarily consists of the genus *Bifidobacterium* [26].

The Firmicutes/Bacteroidetes ratio (F/B) has been shown to influence gut homeostasis and the development of different diseases [27,28]. This F/B ratio can be impacted by changes in the number of certain species of *Firmicutes* or *Bacteroidetes*. For example, a higher F/B ratio is triggered by a higher number of *Firmicutes*, which is frequently linked to obesity [27,29]. On the other hand, a higher *Bacteroidetes* count decreases the F/B ratio, which is associated with IBD [9,27]. Therefore, the balance of the gut ecosystem is critical for sustaining the physiological functions and the health of the human body.

There are multiple ways that gut microbiome might influence biological processes. For example, they play an important role in energy and nutritional extraction from food because of their varied metabolic genes, which supply unique distinct enzymes and biochemical pathways [30]. Furthermore, they play a critical role in the synthesis of bioactive substances such as vitamins, amino acids, and lipids. Additionally, in terms of the immune system, the gut microbiome plays a vital role in the development of the intestinal mucosa and immune system in addition to synthesizing antimicrobial compounds that protect the host from surrounding pathogens [31].

### 2.2. Effect of Diet on Gut Microbiome Community

Diet plays a crucial role in shaping the gut microbiota and influences the types of microbial metabolites produced, either favorably or unfavorably [32]. In newborns, nutritional intake significantly impacts the initial development of the gut microbiome, which adapts in response to the nutrients provided. In early life, the microbiome harbors genes that assist in digesting oligosaccharides found in breast milk. As solid foods are introduced, genes involved in processing polysaccharides and synthesizing vitamins become more prominent [33]. Beyond infancy, dietary habits continue to be one of the primary factors determining the composition and diversity of the gut microbiota [34].

Various dietary patterns lead to different microbial shifts. For instance, plant-based diets often promote the growth of *Firmicutes* and *Bacteroidetes* species [35]. Adequate fiber intake supports the maintenance of the gut’s mucosal lining [36]. Conversely, diets rich in fats and proteins are associated with a rise in Bacteroidetes such as *Bilophila*, *Bacteroides*, and *Alistipes*—microbes that are resilient to bile acids and thrive when *Firmicutes* decline [37]. Prolonged consumption of such high-fat, high-protein diets may impair immune function and raise the risk of infections and metabolic diseases [38,39]. Additionally, diets high in saturated fats and processed sugars, combined with sedentary lifestyles, are known to disrupt the gut microbial balance, contributing to poor glycemic control [40].

When the intake of simple sugars surpasses the absorption capacity of the small intestine, excess carbohydrates reach the colon, providing a rich substrate for bacterial proliferation [41]. High-carbohydrate diets have also been linked to reduced microbial diversity and an increase in saccharolytic bacteria in feces—species not typically dominant in the distal colon [42,43]. Research suggests that sucrose-rich diets may disrupt the gastrointestinal role of *Lactobacillus plantarum*, leading to imbalanced microbial communities. This dysbiosis is characterized by a marked decline in *Lactobacillus*, *Sphingomonas*, and *Klebsiella*, alongside a rise in *Clostridium* and *Bacilli* populations [44,45].

### 2.3. The History of NNS

NNS were originally offered in 1877 as an alternative to traditional table sugar. NNS provides few to no calories, while nutritive sweeteners (NS), such as traditional table sugar, have a caloric load [46]. Both NNS and NS improve food flavor and texture, with a focus on sweetness [47]. However, NNS are up to 20,000 times sweeter than NS (Figure 2) [48]. NNS are regarded as a safe and economical substitute for sugar-sweetened beverages (SSBs), especially among overweight and obese individuals, to lower caloric consumption and improve weight control [49,50]. Additionally, because of their remarkable sweetness, consumers add significantly less amount of NNS to foods, subsequently lowering caloric intake. Diabetic and obese individuals utilize NNS to lower their carbohydrate and caloric intake to control their blood sugar levels and lose weight [46].

Up till now, the FDA has approved six NNS as food additives [51]. These include advantame, neotame, sucralose, saccharin, aspartame, and acesulfame K (Table 1, Figure 3) [52]. In addition, stevioside and rebaudioside A, two naturally derived sweeteners from stevia (*Stevia rebaudiana*) extracts and monk fruit extract (*Siraitia grosvenorii*), respectively, are FDA-recognized as safe (GRAS) zero-calorie sweeteners [53,54].

## 3. Interactions Between NNS and Gut Microbiome

The gut microbiome represents a complex community of microorganisms that interact with dietary components before, during, and after the digestive process. These microbes play a key role in fermenting indigestible substances like dietary fiber, thereby promoting the extraction of energy from compounds the human host cannot digest. Additionally, they contribute to the synthesis of essential nutrients, including various vitamins, making them available for absorption [55]. Notably, emerging research suggests that the gut microbiome can metabolize non-nutritive sweeteners (NNS) in ways comparable to how it processes xenobiotics, such as pharmaceutical drugs [56,57,58]. This interaction has garnered attention because of its potential implications for metabolic conditions, including insulin resistance, obesity, and systemic inflammation [59].

Multiple studies examining the physiological effects of NNS have identified the gut microbiota as a key mediator, raising questions about the long-term safety and host-microbiome dynamics of these compounds [3,60,61,62,63,64]. With advancements in sequencing technologies and computational tools—alongside decreasing costs associated with microbiome analysis—interest in understanding the gut microbiome’s role in human health has expanded significantly [65]. In particular, investigating how widely consumed NNS such as acesulfame K, aspartame, sucralose, and saccharin interact with gut microbes is essential to fully assess their health-related consequences.

### 3.1. Acesulfame K (Ace-K)

The effects of Ace-K on gut microbiome and fecal metabolic profiles were investigated using mouse models. Bian et al. [66] investigated the effects of Ace-K on the gut microbiome and the alterations in fecal metabolic patterns using 16S rRNA sequencing and gas chromatography-mass spectrometry (GC-MS) metabolomics. Following a 4-week therapy, they discovered that Ace-K consumption altered the gut microbiome of CD-1 mice. Following a 4-week treatment, they discovered that Ace-K consumption altered the gut microbiome of CD-1 mice. Ace-K treatment raised *Bacteroides*, *Anaerostipes*, and *Sutterella* in male mice, while Ace-K lowered *Lactobacillus* and *Clostridium* and increased *Mucispirillum* in female mice. They also found that ace-K boosted the body weight gain of male mice but not female mice (Table 2).

Olivier-Van et al. [67] discovered that NNS use during pregnancy and lactation may have harmful consequences on newborn metabolism. The authors studied the effect of 0.1 or 0.2 mg Ace-K together with sucralose in pregnant mice. They demonstrated a significant shift in microbiome profiling in newborns (an increase in *Firmicutes* and a decrease in *Akkermansia muciniphila*), which may be associated with metabolic disease and obesity. In addition, they reported a drastic metabolic change in newborns, suggesting extensive downregulation of hepatic detoxification mechanisms, due to significant downregulation in bile acid-CoA: amino acid N-acetyl transferase Baat, acyl-CoA synthetase Acsl3, glycine modifying enzyme Gnmt, and glutathione S-transferase Gstm5, as well as changes in bacterial metabolites (Table 2).

Hanawa et al. [68] studied the effect of Ace-K on the gut microbiome of C57BL/6J mice and demonstrated that Ace-K can induce dysbiosis and intestinal injury with enhanced lymphocyte migration to intestinal mucosa. They also demonstrated a reduced prevalence of *Clostridiaceae*, *Ruminococcaceae*, and *Lachnospiraceae* in the gut microbiome (Table 2).

On the other hand, other studies did not find any significant changes in gut microbiome upon Ace-K intake [69,70,71] (Table 2).

### 3.2. Aspartame

Palmnäs et al. [63] investigated the impact of chronic low-dose aspartame ingestion on microbiome profile and metabolic parameters in a diet-induced obese rat model. Aspartame was supplied in conjunction with a high-fat (60% kcal fat) diet in this study. The authors discovered that aspartame increased total bacteria, Enterobacteriaceae, and *Clostridium leptum* levels. Furthermore, they found that aspartame decreased caloric intake, weight gain, and body composition in a high-fat diet group, but it also raised fasting glucose levels and prevented insulin-stimulated glucose elimination. In line with this, Nettleton et al. [72] investigated the effects of maternal low-dose aspartame consumption on gut microbiome in obese mothers and their newborns. In this study, pregnant female rats were fed a high-fat/sucrose (HFSD) diet with or without 5–7 mg/kg body weight aspartame. The authors found that mothers had a higher prevalence of *Akkermansia muciniphila* and Enterobacteriaceae than their newborns. The aspartame group had lower amounts of Enterococcaceae, *Enterococcus*, and *Parasutterella*, as well as higher levels of *Clostridium cluster IV*, in the cecal matter from newborns. Furthermore, there were higher levels of Porphyromonadaceae in the guts of both male and female newborns. Recently, Suez et al. [15] investigated the effects of aspartame, saccharin, and sucralose on the fecal and oral microbiome of adult humans. The authors discovered that aspartame and other tested NNS significantly impacted the human gut microbiome and their functions when compared to the control groups. However, it was not clear which types of bacteria increased or decreased. The oral microbiome was similarly impacted by aspartame, which decreased the prevalence of *Porphyromonas* and *Prevotella nanceiensis* [15].

On the other hand, Frankenfeld et al. [71] found no significant difference in bacterial abundance in adult humans who consumed aspartame and acesulfame-K throughout their cross-sectional study. However, they detected a reduced diversity of bacteria (from 24 to 7 phyla). Additionally, Ahmed et al. [73] discovered that daily repeated administration of pure aspartame or sucralose at levels corresponding to normal ADI had little influence on the gut microbiome composition or short-chain fatty acid (SCFA) production.

### 3.3. Sucralose

Given that sucralose itself is not metabolized by gut microbes, studies reporting effects on the intestinal microbiota should clarify whether they utilized pure sucralose or commercial preparations. This distinction is important, as commercial products generally contain only about 1% sucralose, with the remaining 99% composed of carriers such as maltodextrins [74].

In this regard, Rodriguez-Palacios et al. [2] investigated the effects of 6-week Splenda^®^ (sucralose and maltodextrin, 1:99 w/w) supplementation on Crohn’s disease-like ileitis severity and changes in the gut microbiome in SAMP1/YitFc (SAMP) mice. They found that Splenda^®^ induced dysbiosis in all mice by expanding Proteobacteria, as well as *E. coli* overgrowth by increasing bacterial penetration into the ileal lamina propria of SAMP mice. Additionally, Bian et al. [3] investigated the structural and functional effects of sucralose on the gut microbiome and associated inflammation in the host using C57BL/6 male mice model. After three and six months of sucralose treatment, the authors observed a disturbance in fecal metabolites and an enrichment of gut microbiome pro-inflammatory genes. Additionally, following six months of treatment, they noted a rise in the expression of hepatic pro-inflammatory genes. These data imply that sucralose at a dose level similar to the human ADI may raise the risk of inflammation by altering the gut microbiome [3].

In another study, Wang et al. [75] examined the effect of sucralose on the gut microbiome when combined with normal diet and high-fat diet in mice models. The authors noted that, in mice given sucralose with a normal diet, there was an increase in Firmicutes, which additionally expanded when sucralose was given with the high-fat diet. A dysbiotic impact was also detected in the newborns of mothers treated for 6 weeks with sucralose in conjugation with Ace-K. In these newborns, the intestinal levels of *Firmicutes* doubled, particularly in the Clostridiales families Lachnospiraceae and Ruminococcaceae [67]. However, according to the Uebanso et al. study [69], mice who received pure sucralose (14.2 mg/kg bw/day) for 8 weeks had a lower relative concentration of *Clostridium cluster XIVa* in their fecal microbiota.

Dai et al. [76] investigated the effect of maternal sucralose consumption on a newborn’s probability of developing hepatic steatosis in adulthood. They discovered that maternal sucralose intake alters the makeup and diversity of the gut microbiome, reduces butyrate-producing bacteria and butyrate, interferes with intestinal development, induces low-grade inflammation, and disrupts gut barrier function in 3-week-old newborns. Furthermore, microbial analysis in adulthood confirmed that dysbiosis caused by maternal sucralose consumption is long-lasting and may increase the risk of developing steatosis in adulthood. Finally, maternal sucralose intake triggered HFD-induced hepatic steatosis in adulthood [76].

Recently, Zheng et al. [77] investigated the effect of low-dose sucralose on the gut microbiome. In this study, mice were given 0.0003, 0.003, 0.03, and 0.3 mg/mL of sucralose for 16 weeks. Notably, the 0.3 mg/mL of sucralose was equivalent to the ADI (5 mg/kg BW/day). The authors reported that sucralose significantly changed the composition of the gut microbiome, especially at doses of 0.3 mg/mL, which led to an increase in the number of possible pathogens in the jejunum, ileum, and colon, including *Tenacibaculum*, *Ruegeria*, and *Staphylococcus*. Sucralose also raised *Allobaculum* at this dose, which was found to have a positive association with diabetes. The cecum also had lower levels of *Lachnoclostridium* and *Lachnospiraceae* than the controls [77].

Regarding human trials, Thomson et al. [78] conducted the first study that evaluated the effects of sucralose consumption on adult humans. In this study, 34 healthy male volunteers were given 780 mg/day of sucralose for 7 days. The authors found no changes in glycemic control, insulin resistance, or intestinal microbiome at the phylum level in the sucralose-treated group.

### 3.4. Saccharin

In 2014, Suez et al. [60] examined saccharin’s impacts on mice’s gut microbiome over 5 weeks, using 5 mg/kg bw of saccharin and sequencing their *16S* rRNA gene. Compared to water and glucose control groups, saccharin led to a unique gut microbiome. Saccharin consumption in mice was associated with gut dysbiosis, with an increase in *Bacteroides* and *Clostridiales* and a reduction in *Lactobacillus reuteri*. The study also evaluated saccharin effects in 7 healthy adults, each of whom consumed 5 mg/kg bw/day of saccharin for 6 days, aligning with the FDA’s maximum acceptable daily intake. Four participants developed impaired glucose tolerance (responders), while three did not (non-responders), indicating that saccharin may affect individuals differently based on gut microbiome composition. In responders, the study found a significant increase in *Bacteroides fragilis* and *Weissella cibaria* as well as a 10-fold decrease in *Candidatus arthromitus*, whereas non-responders showed minimal microbiome changes. Additionally, to assess whether saccharin-induced microbiome changes could cause glucose intolerance, stool samples from responders and non-responders, taken before and after saccharin intake, were transplanted into germ-free mice. Only the mice that received stool from saccharin responders after treatment developed glucose intolerance. This result suggests that changes in the gut microbiome from saccharin might be responsible for these glucose issues [60].

Furthermore, Bian et al. [3] investigated saccharin’s effects on gut microbiome in male C57BL/6J mice by providing 0.3 mg saccharin per mL in drinking water over 6 months. Using 16S rRNA sequencing, they observed significant microbiome changes, with 11 bacterial genera altered at 3 and 6 months. Specifically, *Sporosarcina*, *Jeotgalicoccus*, *Akkermansia*, *Oscillospira*, and *Corynebacterium* increased at 3 months, while *Corynebacterium*, *Roseburia*, and *Turicibacter* increased at 6 months; in contrast, *Anaerostipes* and *Ruminococcus* declined at 3 months, and *Ruminococcus*, *Adlercreutzia*, and *Dorea* declined at 6 months. Additionally, hepatic pro-inflammatory genes like nitric oxide synthase and TNF-α were elevated after 6 months [3].

Additionally, two veterinary studies reported the effects of saccharine on microbiome composition of piglets [79,80]. One study found that the addition of 0.015% (*w/w*) saccharine to basal diet for two weeks increased the cecal *Lactobacillus* populations, specifically *Lactobacillus OTU4228* [79]. Another study reported that saccharine led to a considerable increase in the abundance of *Lactobacillus* population, specifically *Lactobacillus OTU4228* [80].

On the other side, other veterinarians as well as human studies reported that saccharine had no significant effect on gut microbiome [70,81,82].

### 3.5. Neotame

Chi et al. [83] investigated the impact of neotame exposure on the gut microbiome in CD-1 mice over four weeks. They observed an increase in two genera from the phylum Bacteroidetes—*Bacteroides* and an undefined genus within the S24-7 group—while noting a significant reduction in three genera of the family Ruminococcaceae (*Oscillospira*, *Ruminococcus*, and one undefined genus) and five genera of Lachnospiraceae, including *Blautia*, *Dorea*, *Ruminococcus*, and two undefined genera. This remains the sole study examining neotame’s effects on the gut microbiome.

### 3.6. Stevia

Singh et al. [84] investigated the impact of 12 weeks of stevia consumption on the human gut microbiota. The study found no significant changes in the microbial composition of participants’ gut microbiota, based on the analysis of fecal samples before and after the consumption period. This supports the idea that stevia may have a neutral effect on gut health and does not disrupt the gut ecosystem in the context of a typical diet.

Katsi et al. [85] explored the effects of stevia on gut bacteria, emphasizing its potential impact on microbial communities. The review suggests that while stevia has shown potential as a prebiotic, promoting beneficial bacteria such as *Lactobacillus* and *Bifidobacterium*, it does not consistently alter the microbiome across all studies. The study also highlighted that stevia’s influence on gut bacteria may vary based on dosage, the individual’s health status, and the type of stevia used. Overall, stevia consumption seems to have a minimal, neutral effect on gut microbiota composition. The study by Sakr and Massoud [86] highlighted the prebiotic potential of stevia sweeteners, particularly in supporting the growth of beneficial gut bacteria such as *Lactobacillus* and *Bifidobacterium*. These bacteria are known for their role in improving gut health, enhancing digestion, and supporting the immune system.

On the other hand, several studies found that stevia may impact the composition of the gut microbiota. For instance, Becker et al. [87] investigated the effects of stevia supplementation on glucose tolerance and gut microbiota in mice. They found that stevia supplementation did not reverse high-fat diet-induced glucose intolerance or microbiota changes. It also showed that stevia caused similar alterations to the gut microbiota as saccharin when administered alongside a high-fat diet. Another study by Deniņa et al. [88] evaluated the effects of stevia glycosides (stevioside and rebaudioside A) on *Lactobacillus reuteri* strains, which are beneficial gut bacteria that help maintain gut health by producing lactic and acetic acid. The results showed that both glycosides inhibited the growth of these strains in a concentration-dependent manner. The effects were strain-specific, with notable changes in the production of lactic and acetic acids. These findings suggest that stevia glycosides may alter the growth of certain gut bacteria, impacting digestive health. Additionally, Nettleton et al. [72] investigated the impact of low-dose aspartame and stevia, consumed along with an obesogenic diet, on metabolism, gut microbiota, and the mesolimbic reward system in rat dams and their offspring. The results show that both stevia and aspartame alter the gut microbiota, with specific changes observed in the diversity and composition of microbial populations. However, the study emphasizes the need for further exploration to understand how these AS might affect long-term health outcomes, including metabolic disorders. In a further study, Nettleton et al. [89] found that stevia consumption perturbed the gut microbiota, including reductions in diversity and alterations in specific bacterial populations, such as *Lachnospiraceae* and *Ruminococcaceae*. These changes were associated with modifications in the mesolimbic dopamine pathway, potentially influencing reward-related behaviors. The findings highlight the interplay between sweetener consumption, gut microbiota, and neurological functions.

**Table 2 diseases-13-00115-t002:** Effect of the NNS on gut microbiome.

NNS	Study Type/Model	Dose and Exposure Time	Outcome(s)	Ref.
Acesulfame K (Ace-K)	Mice	37.5 mg/kg bw/day for 4 weeks	1. Altered the gut microbiome: In male mice ↑ *Bacteroides*, *Anaerostipes*, and *Sutterella* In female mice ↓ *Lactobacillus* and *Clostridium* ↑ *Mucispirillum* 2. Increased body weight gain of male mice.	[66]
	Female mice	0.1 or 0.2 mg Ace-K together with sucralose for 6 weeks	1. Altered the gut microbiome of newborns: ↑ *firmicutes* ↓ *Akkermansia muciniphila* 2. Drastic metabolic change in newborns	[67]
	C57BL/6J mice	150 mg/kg bw/day for 8 weeks	1. Altered the gut microbiome ↓ *Clostridiaceae*, *Ruminococcaceae*, and *Lachnospiraceae* 2. Intestinal injury with enhanced lymphocyte migration to the intestinal mucosa.	[68]
	Male mice	15 mg/kg bw/day for 8 weeks	No significant effects	[69]
	Mice	40 or 120 mg/kg bw/day for 4 weeks	No significant effects	[70]
	Human	1.7–33.2 mg/kg bw/day for 4 days	No significant effects	[71]
Aspartame	Obese rat	High-fat diet + 5–7 mg/kg bw/day aspartame (in drinking water) for 8 weeks	1. Altered the gut microbiome ↑ total bacteria, Enterobacteriaceae, and *Clostridium leptum* 2. Reduced caloric intake, weight gain, and body composition in the high-fat diet group. 3. Increased fasting glucose levels and impaired insulin-stimulated glucose elimination in the high-fat diet group.	[63]
	Pregnant rats and their newborns	High fat/sucrose (HFS) diet + 5–7 mg/kg bw/day aspartame for 18 weeks	1. ↑ *Akkermansia muciniphila* and Enterobacteriaceae in mother than their newborns. 2. ↓ Enterococcaceae, *Enterococcus*, and *Parasutterella*, as well as ↑ *Clostridium cluster IV*, in cecal matter from newborns. 3. ↑ Porphyromonadaceae in the guts of both male and female newborns.	[72]
	Human	0.24 g/day	1. Altered the gut microbiome. 2. ↓ *Porphyromonas* and *Prevotella nanceiensis* in oral microbiome.	[15]
	Human	62.7 mg/day for 4 days	1. No significant difference in bacterial abundance. 2. ↓ diversity of bacteria (from 24 to 7 phyla).	[71]
	Human	0.425 g/day for 2 weeks	Little influence on gut microbiome composition or SCFA production	[73]
Sucralose	SAMP1/YitFc mice	3.5 mg/mL of Splenda^®^ (sucralose maltodextrin, 1:99 *w/w*)	1. Splenda^®^ induced dysbiosis in all mice: ↑ Proteobacteria ↑ *E. coli* overgrowth by increasing bacterial penetration into the ileal lamina propria of SAMP mice	[2]
	C57BL/6J male mice	5 mg/kg bw/day for 3 and 6 months	1. Alterations in bacterial genera After 3 months ↑ *Ruminococcus* ↓ *Lachnospiraceae*, *Dehalobacteriaceae*, *Anaerostipes*, *Staphylococcus*, *Peptostreptococcaceae*, and *Bacillus*. After 6 months ↑ *Akkermansia*, *Turicibacter*, *Roseburia*, *Clostridiaceae*, and *Christensenellaceae* ↓ *Streptococcus*, *Lachnospiraceae*, *Dehalobacteriaceae*, and *Erysipelotrichaceae*. 2. Following six months, ↑ expression of hepatic pro-inflammatory genes	[3]
	Mice	3.3 mg/kg bw/day sucralose + normal diet 1.5 mg/kg bw/day sucralose + high-fat diet	↑ Firmicutes in mice given sucralose + normal diet ↑ Further elevation of Firmicutes in mice given sucralose + high-fat diet	[75]
	Pregnant mice	Sucralose + 0.1- 0.2 mg Ace-K for 6 weeks	↑ Firmicutes and ↓ *Akkermansia muciniphila* in newborns.	[67]
	Male mice	15 mg/kg bw/day sucralose for 8 weeks	↓ *Clostridium cluster XIVa* in the fecal microbiota	[69]
	Pregnant mice	Sucralose solution of 0.1 mg/mL for 6 weeks	1. Maternal sucralose intake led to (in 3-weeks-old newborns): altered composition and diversity of the gut microbiome reduced butyrate-producing bacteria and butyrate, disrupted gut barrier function 2. Maternal sucralose intake enhanced HFD-induced hepatic steatosis in adulthood.	[76]
	Mice	0.0003–0.3 mg/mL of sucralose	↑ *Tenacibaculum*, *Ruegeria*, and *Staphylococcus* in the jejunum, ileum, and colon (mainly at 0.3 mg/mL dose). ↑ *Allobaculum* (at 0.3 mg/mL dose), which was found to have a positive association with diabetes. ↓ *Lachnoclostridium* and *Lachnospiraceae* present in cecum	[77]
	Human	780 mg/day of sucralose for 7 days	No changes in glycemic control, insulin resistance, or intestinal microbiome at the phylum level in the sucralose-treated group	[78]
	Human	0.136 g/day sucralose for 2 weeks	No significant effects	[73]
	Human	48 mg/day sucralose for 10 weeks	1. Gut microbiome dysbiosis 3-fold increase in *Blautia coccoides* ↓ *Lactobacillus acidophilus* 2. ↑ serum insulin and the area under the glucose curve.	[16]
Saccharin	Mice	5 mg/kg bw of saccharin for 5 weeks	1. Gut microbiome dysbiosis ↑ *Bacteroides* and *Clostridiales* ↓ *Lactobacillus reuteri* 2. Impaired glucose tolerance	[60]
	Human	5 mg/kg bw/day of saccharin for 6 days	↑ *Bacteroides fragilis* and *Weissella cibaria* A 10-fold decrease in *Candidatus arthromitus*	[60]
	C57BL/6J male mice	0.3 mg saccharin per mL in drinking water over 6 months	1. Alterations in bacterial genera After 3 months ↑ *Sporosarcina*, *Jeotgalicoccus*, *Akkermansia*, *Oscillospira*, and *Corynebacterium*; ↓ *Anaerostipes* and *Ruminococcus* After 6 months ↑ *Corynebacterium*, *Roseburia*, and *Turicibacter* ↓ *Ruminococcus*, *Adlercreutzia*, and *Dorea* 2. Following six months, ↑ expression of hepatic pro-inflammatory genes	[3]
	Human	180 mg/day saccharin + 5820 mg/day glucose for 2 weeks	1. Altered the gut microbiome. 2. ↓ *Porphyromonas* and *Prevotella nanceiensis* in oral microbiome.	[15]
	Piglets	Basal diet + 0.015% (*w/w*) saccharin for 2 weeks	↑ *Lactobacillus* population, particularly *Lactobacillus OTU4228*	[79,80]
	Dog	0.02% of saccharin + eugenol for 10 days	No significant effects	[81]
	Mice	250 mg/kg bw/day saccharin	No significant effects	[82]
	Mice	20–100 mg/kg bw/day saccharin for 4 weeks	No significant effects	[70]
	Human	400 mg/day saccharin for 2 weeks	No significant effects	[82]
Neotame	Mice	0.75 mg/kg bw/day neotame for 4 weeks	1. ↑ two genera from the phylum Bacteroidetes—*Bacteroides* and an undefined genus in S24-7. 2. ↓ in three genera of the family Ruminococcaceae (*Oscillospira*, *Ruminococcus*, and one undefined genus) and five genera of Lachnospiraceae, including *Blautia*, *Dorea*, *Ruminococcus*, and two undefined genera.	[83]

The discussion highlights the inconsistencies in studies examining the impact of NNS on gut microbiome. Variability stems from differences in study design, species metabolism, and undefined concepts of dysbiosis or eubiosis. While some human trials suggest dysbiotic effects, others report negligible impact, with results influenced by participants’ diet, lifestyle, and environment. This underscores that NNS consumption is one among many factors shaping the microbiome. More focused studies are needed to determine the specific effects of NNS at regulatory-approved doses and their broader implications.

## 4. Possible Mechanisms for Interactions Between Gut Microbiome and NNS

Recent research has explored how NNS influences host physiology via the gut microbiome. One proposed mechanism involves interactions between NNS and gut taste receptors, which may regulate glucose absorption and homeostasis [90]. These interactions might trigger physiological responses similar to those induced by sugar sweeteners, such as cephalic phase insulin release [91]. However, the lack of consistent evidence for insulin release in response to NNS complicates this hypothesis. Further investigation is needed to clarify these mechanisms and their implications for host physiology [64].

Another possible mechanism through which NNS influence gut physiology involves their direct action on the gut epithelium. NNS may influence gut physiology by acting directly on the gut epithelium, thereby affecting mucin production and gut barrier function [92,93,94], which regulate microbiome composition and metabolism [92,95]. These effects can deplete mucin-dependent bacteria like *Akkermansia* [96]. Additionally, some NNS may have “antibiotic-like” effects, damaging bacterial membranes and altering permeability [97]. Conversely, the metabolic byproducts of these compounds, from either the host [98,99,100] or gut bacteria [56,101,102], may impact the microbiome and gut barrier, resulting in shifts in microbial populations.

Although only a small amount of NNS reaches the intestine, the gut microbiome can metabolize them, leading to various biological effects (Figure 4). Research suggests that gut microbiome can metabolize NNS, leading to various biological outcomes. Some bacteria use NNS as a carbon source, altering metabolic activity and SCFAs [3,60] production, such as acetate, propionate, and butyrate, which influence glucose metabolism and provide anti-inflammatory effects [103,104,105]. Additionally, NNS may disrupt the gut microbiome by reducing beneficial bacteria (e.g., *Bifidobacterium*, *Lactobacillus*) [106,107] and promoting harmful strains (e.g., *Clostridium difficile*, *E. coli*), potentially triggering inflammation, gut dysbiosis, and conditions like IBD [9,108,109]. Additionally, NNS can modulate gut hormone release and bile acid levels, affecting motility, nutrient absorption, and overall microbiome composition [110,111].

Beyond these pathways, the gut microbiota plays a critical role in xenobiotic metabolism, including the transformation of NNS [112]. The microbial community harbors enzymes such as glucuronidases, sulfatases, carboxylesterases, and amidases, capable of chemically modifying xenobiotics into active or inactive metabolites. For example, β-glucuronidase can deconjugate compounds excreted via bile, leading to reactivation and potential local effects. These enzymatic processes can influence the toxicity, bioavailability, and physiological impact of artificial sweeteners [113].

While these mechanisms provide a framework for understanding how NNS impact host-microbiome dynamics, many questions remain. Notably, the molecular pathways linking NNS-induced microbial changes to host metabolic and immune responses require further investigation. Future research should employ integrated omics approaches—such as metabolomics to profile bacterial metabolites, transcriptomics to assess gene expression changes, and metagenomics to explore microbial community shifts. These tools can unravel the complex crosstalk between NNS, gut bacteria, and host physiology, offering mechanistic insights into how dietary choices shape health outcomes.

## 5. Recommendations

In light of recent recommendations from the World Health Organization (WHO) [114], the use of non-nutritive sweeteners (NNS) should not be relied upon as a primary strategy for weight control or the prevention of non-communicable diseases. Evidence indicates that NNS consumption does not lead to significant long-term weight loss and may pose potential metabolic health risks, including an increased likelihood of Type 2 diabetes and cardiovascular diseases. Instead, individuals should focus on reducing free sugar intake and adopting a holistic dietary approach that prioritizes whole foods, fruits, vegetables, and balanced nutrition. This shift promotes not only effective weight management but also improved overall health outcomes.

Given the existing evidence, public health guidelines should provide clearer recommendations on NNS consumption for different population groups. For individuals with diabetes, NNS may serve as an alternative to sugar in managing blood glucose levels; however, careful selection and moderation are necessary, as some sweeteners may still influence metabolic pathways. Healthcare professionals should guide patients toward informed decisions based on individual metabolic responses. In the case of obese individuals, while NNS can assist in lowering calorie intake, their long-term efficacy in weight management remains uncertain. Emphasizing nutrient-dense whole foods and sustainable behavioral changes should take precedence over artificial substitutes. For the general healthy population, limiting both sugar and NNS intake is advisable. Encouraging a diet rich in natural, unprocessed foods can help maintain metabolic and gut health while reducing dependence on artificial sweeteners.

To further support reduced reliance on NNS, dietary modifications should be emphasized as part of public health strategies. Increasing dietary fiber intake through whole grains, legumes, and vegetables can aid in appetite regulation and blood sugar control, reducing the desire for excessive sweetness. A gradual reduction in refined sugar consumption can help shift taste preferences toward naturally sweet foods like fruits, fostering long-term metabolic stability. Additionally, education and awareness campaigns promoting mindful eating, portion control, and the benefits of a balanced diet without excessive sweeteners can empower individuals to make healthier choices.

## 6. Conclusions and Future Perspectives

Artificial sweeteners represent a double-edged sword in their interaction with the gut microbiome. On the one hand, they provide a low-calorie alternative to sugar, aiding in weight management and glycemic control. On the other hand, accumulating evidence suggests that AS may disrupt the delicate balance of the gut microbiome, inducing dysbiosis, inflammation, and metabolic alterations. The observed reduction in beneficial bacteria, coupled with an increase in potentially pathogenic strains, highlights the need for caution in AS consumption, particularly among individuals with compromised gut health.

Despite numerous studies exploring the effects of NNS on the gut microbiome, results remain conflicting. While some investigations report minimal or no significant changes in microbial composition, others demonstrate notable alterations. These inconsistencies are likely due, in part, to differences in study design—including variability in dosage, exposure duration, and the choice of experimental models (in vivo vs. in vitro). Additionally, a lack of clarity regarding whether NNS were administered as pure compounds or commercial formulations complicates interpretation, as additives may exert independent effects on microbial communities.

To address these gaps, future research should prioritize the standardization of experimental protocols. Employing consistent dosages, uniform exposure times, and comparable experimental systems will enhance the reproducibility and comparability of findings. Additionally, clearly distinguishing between pure compounds and commercially available formulations will help isolate the specific effects of NNS and reduce confounding variables.

Another critical aspect requiring attention is inter-individual variability in response to NNS. Factors such as genetic background, habitual dietary intake, existing microbiota composition, and lifestyle choices may modulate the effects of NNS on the gut microbiome. To account for this, future studies should incorporate stratified analyses and recruit diverse, well-characterized clinical populations. This approach will help identify population subgroups that are more susceptible to NNS-induced microbiota disturbances and will support the development of more tailored dietary guidelines.

Long-term, well-controlled studies are essential to fully elucidate the impacts of NNS on gut health and overall human well-being. By addressing both methodological variability and individual differences, future research can pave the way for evidence-based recommendations that prioritize both metabolic benefits and microbiome integrity—aligning dietary strategies with long-term wellness goals.

## Figures and Tables

**Figure 1 diseases-13-00115-f001:**
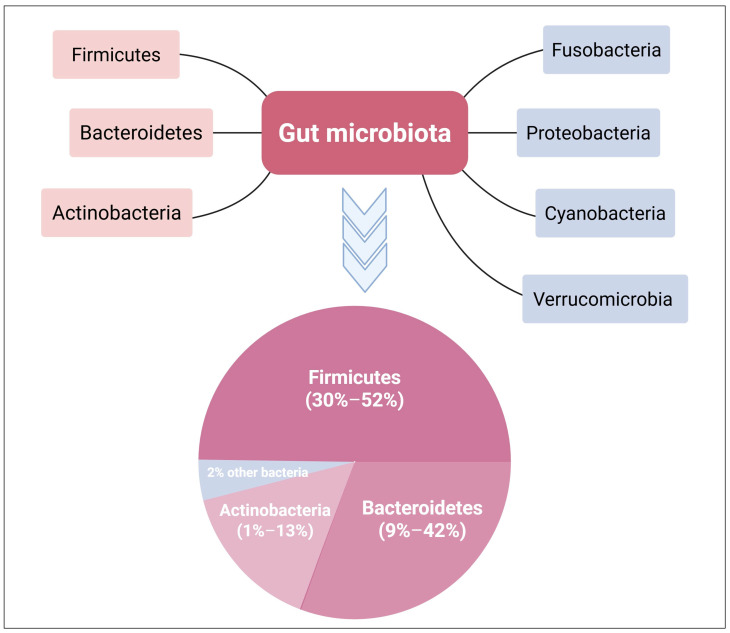
Phylum distribution in the gut microbiome. The dominant bacterial phyla in the gut microbiome are *Bacteroidetes*, *Firmicutes*, and *Actinobacteria*, comprising around 98% of the total population. *Bacteroidetes* and *Firmicutes* together account for over 90% of the microbial phyla in the GIT. Created with BioRender.com.

**Figure 2 diseases-13-00115-f002:**
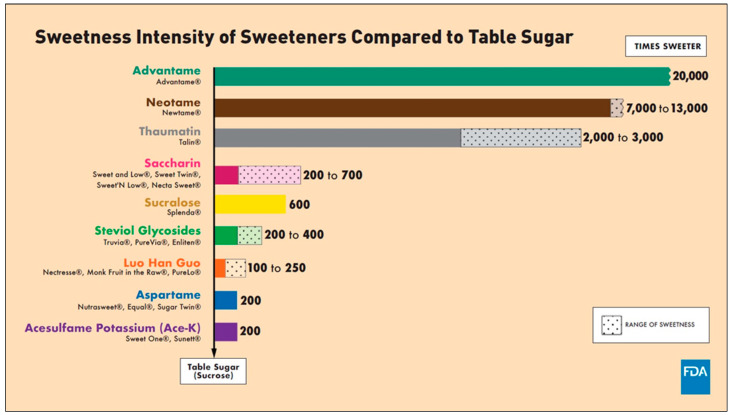
Sweetness intensity of NNS compared to table sugar, as proved by FDA [48]. (accessed on 1 March 2025).

**Figure 3 diseases-13-00115-f003:**
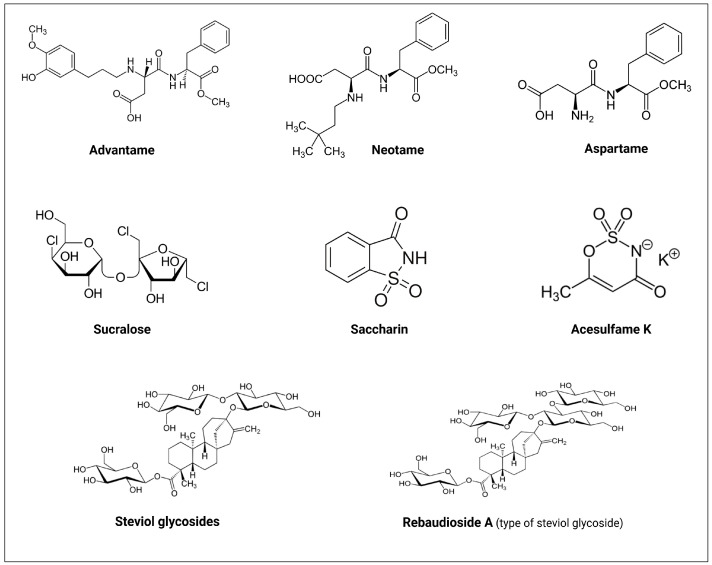
Molecular structures of the FDA approved non-nutritive sweeteners (NNS). Created with BioRender.com.

**Figure 4 diseases-13-00115-f004:**
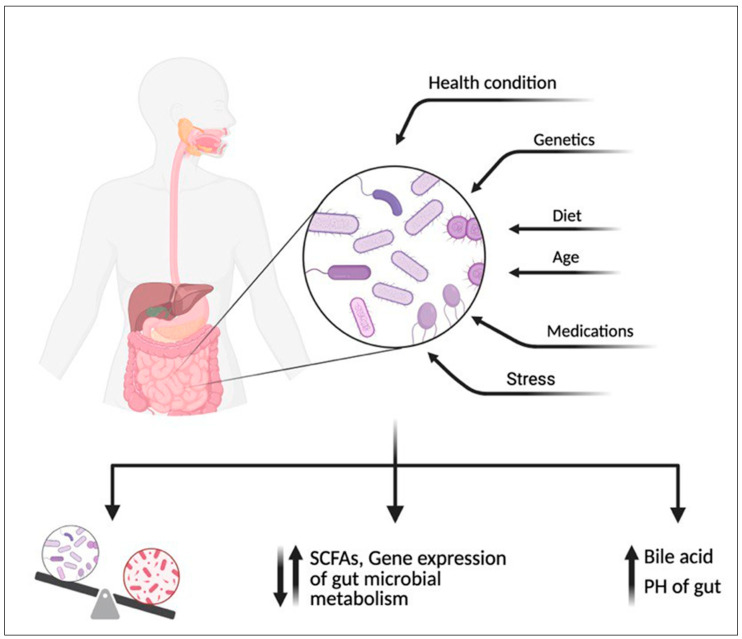
An overview of the documented effects of non-nutritive sweeteners (NNS) on human and gut microbiome metabolism highlights their dual impact. NNS can alter the composition and activity of the gut microbiome, with some strains altering the metabolism of short-chain fatty acids (SCFAs) like acetate, propionate, and butyrate, influencing glucose regulation and inflammation, increasing the PH of the gut or increasing bile acid secretion. In humans, these metabolic changes may disrupt gut health. Created with BioRender.com.

**Table 1 diseases-13-00115-t001:** FDA-approved NNS [52].

NNS	FDA Approval Year	Acceptable Daily Intake (ADI)	Daily Sweetener Packet Limit
Advantame	2014	32.8 mg/kg bw/d	4920
Neotame	2002	0.3 mg/kg bw/d	23
Sucralose	1998	5 mg/kg bw/d	23
Saccharin	1977	15 mg/kg bw/d	45
Aspartame	1981	50 mg/kg bw/d	75
Acesulfame K	1988	15 mg/kg bw/d	23
Steviol glycosides	2008	4 mg/kg bw/d	9
Rebaudioside A (type of steviol glycoside)	2008	12 mg/kg bw/d	27

## Data Availability

Not applicable.

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
