# Peer review of "Artificial Sweeteners: A Double-Edged Sword for Gut Microbiome"

_diseases, 2025, doi:10.3390/diseases13040115_

Round 1

Reviewer 1 Report

Comments and Suggestions for Authors

The title of this article is “Artificial Sweeteners: A Double-Edged Sword for Gut Microbiome”. This is an interesting topic. However, there are still several parts that need to be revised:
1. The research results mentioned in the paper are inconsistent, partly because of the differences in research design (such as dosage, exposure time, experimental model, etc.). It is suggested that future research should standardize the experimental design as much as possible, including the use of uniform dose, exposure time and experimental model, so as to better compare the results of different studies. In addition, it is recommended to specify whether NNS used is a pure compound or a commercial formula, because additives in commercial formulas may affect the results.
2. It is mentioned in the manuscript that there are individual differences in the influence of NNS on intestinal flora, and some people may be more susceptible to the influence of NNS. It is suggested that future research should consider more individual differences, such as genetic background, eating habits, lifestyle and other factors, in order to better understand the impact of NNS on different groups of people. These differences can be discussed through larger clinical trials or stratified analysis.
3. Although it has mentioned some possible mechanisms (such as the influence of NNS on intestinal epithelial cells, the production of SCFA, etc.), the research on these mechanisms is not deep enough. The authors are suggested to further study how NNS affects the host health by affecting the metabolic activity, immune regulation and intestinal barrier function of intestinal flora. In particular, the authors can explore the molecular mechanism of NNS on intestinal flora and host metabolism through metabonomics, transcriptomics and other technical means.
4. In the Conclusion, the paper mentioned WHO's suggestions, but did not put forward specific public health suggestions on the use of NNS. It is suggested that the public health suggestions based on the existing evidence should be put forward more clearly in the conclusion, especially the guidelines for different groups (such as diabetic patients, obese patients and healthy people). In addition, the authors can explore how to reduce the dependence on NNS through dietary adjustment (such as increasing dietary fiber and reducing refined sugar intake).

Comments on the Quality of English Language

The English could be improved to more clearly express the research.

Author Response

The title of this article is “Artificial Sweeteners: A Double-Edged Sword for Gut Microbiome”. This is an interesting topic. However, there are still several parts that need to be revised:

Comment 1:The research results mentioned in the paper are inconsistent, partly because of the differences in research design (such as dosage, exposure time, experimental model, etc.). It is suggested that future research should standardize the experimental design as much as possible, including the use of uniform dose, exposure time and experimental model, so as to better compare the results of different studies. In addition, it is recommended to specify whether NNS used is a pure compound or a commercial formula, because additives in commercial formulas may affect the results.

Response 1: Thank you for this insightful comment. We agree that heterogeneity in experimental design. In the revised manuscript, we have included a paragraph in the conclusion section emphasizing the need for standardized protocols in future research, particularly with regard to dosage, exposure time, and experimental model. Additionally, we now highlight the importance of clearly distinguishing between pure compounds and commercial formulations of non-nutritive sweeteners (NNS), as the latter may contain additives that confound interpretation of results.

Comment 2: It is mentioned in the manuscript that there are individual differences in the influence of NNS on intestinal flora, and some people may be more susceptible to the influence of NNS. It is suggested that future research should consider more individual differences, such as genetic background, eating habits, lifestyle and other factors, in order to better understand the impact of NNS on different groups of people. These differences can be discussed through larger clinical trials or stratified analysis.

Response 2: Thank you for this valuable comment. In response, we have expanded the "Conclusion and Future Perspective" section to underscore the importance of considering factors such as genetic predisposition, dietary habits, and lifestyle. We also highlight the need for future research to incorporate stratified analyses and larger clinical trials to better elucidate population-specific responses to NNS exposure.

Comment 3: Although it has mentioned some possible mechanisms (such as the influence of NNS on intestinal epithelial cells, the production of SCFA, etc.), the research on these mechanisms is not deep enough. The authors are suggested to further study how NNS affects the host health by affecting the metabolic activity, immune regulation and intestinal barrier function of intestinal flora. In particular, the authors can explore the molecular mechanism of NNS on intestinal flora and host metabolism through metabonomics, transcriptomics and other technical means.

Response 3: We appreciate the reviewer’s insightful suggestion. In response, we have expanded the    " Possible mechanisms for interactions between gut microbiome and NNS" section to further explore potential mechanisms by which NNS may affect host health through modulation of microbial metabolic activity, immune responses, and intestinal barrier integrity. We also highlight the utility of advanced omics-based technologies—such as metabolomics and transcriptomics—as powerful tools to elucidate the molecular pathways involved in these processes.

Comment 4: In the Conclusion, the paper mentioned WHO's suggestions, but did not put forward specific public health suggestions on the use of NNS. It is suggested that the public health suggestions based on the existing evidence should be put forward more clearly in the conclusion, especially the guidelines for different groups (such as diabetic patients, obese patients and healthy people). In addition, the authors can explore how to reduce the dependence on NNS through dietary adjustment (such as increasing dietary fiber and reducing refined sugar intake).

Response 4: Thank you for this valuable suggestion. We have carefully revised the manuscript by incorporating specific public health recommendations based on existing evidence. However, instead of expanding the conclusion with these details, we have integrated them into the Recommendations section for better structural clarity. The revised section now provides clear guidance for different populations, including diabetic patients, obese individuals, and healthy people, while also suggesting dietary adjustments to reduce NNS dependence.

Comments on the Quality of English Language:

The English could be improved to more clearly express the research.

Response: We appreciate the reviewer’s insightful suggestion. Revised and improved.

Reviewer 2 Report

Comments and Suggestions for Authors

Great review, I really enjoyed it. I have a few recommendations below.

1. The description of dysbiosis on page 2 is not complete. I recommend describing it better as you do later on in the review. Dysbiosis entails changes in composition and diversity such as: 1) overgrowth of conditionally pathogenic microbes and metabolites, 2)decrease in conditionally beneficial microbes and metabolites, 3) alterations in host intestinal function due to aberrant colonization or GM- inflammation, etc.

2. On pg.6, please specify what hepatic detoxification mechanisms are changed in the study? There are many hepatic detoxification mechanisms. 

3. It would be nice if you could include the molecular structures of the non-nutritive sweeteners in a reference image.

4. It would be impactful to discuss what is known about xenobiotic metabolism from the gut microbiota. Do these bacteria modify their structure via glucuronidases, carboxylases, amidases, etc.?

5. What is known about how mechanisms of how metabolites of NNS' may exacerbate dysbiosis?

Author Response

Great review, I really enjoyed it. I have a few recommendations below.

Comment 1:The description of dysbiosis on page 2 is not complete. I recommend describing it better as you do later on in the review. Dysbiosis entails changes in composition and diversity such as: 1) overgrowth of conditionally pathogenic microbes and metabolites, 2) decrease in conditionally beneficial microbes and metabolites, 3) alterations in host intestinal function due to aberrant colonization or GM- inflammation, etc.

Response 1: We appreciate the reviewer’s insightful suggestion. In response, we added a conscious and comprehensive description for dysbiosis.

Comment 2:On pg.6, please specify what hepatic detoxification mechanisms are changed in the study? There are many hepatic detoxification mechanisms.

Response 2: We appreciate the reviewer’s insightful note. In response, specific hepatic detoxification mechanisms changed in the study were added.

Comment 3:It would be nice if you could include the molecular structures of the non-nutritive sweeteners in a reference image.

Response 3: We appreciate the reviewer’s insightful suggestion. In response, we added a figure that includes the Molecular structures of the FDA approved non-nutritive sweeteners (NNS).

Comment 4: It would be impactful to discuss what is known about xenobiotic metabolism from the gut microbiota. Do these bacteria modify their structure via glucuronidases, carboxylases, amidases, etc.?

Response 4: We fully agree. A paragraph discussing microbial xenobiotic metabolism—including enzymes such as glucuronidases, carboxylases, and amidases—has been added to Section 3 as suggested.

Comment 5: What is known about how mechanisms of how metabolites of NNS' may exacerbate dysbiosis?

Response 5: We appreciate the reviewer’s insightful suggestion. In response, we added a related paragraph to discuss how mechanisms of how metabolites of NNS' may exacerbate dysbiosis.

Reviewer 3 Report

Comments and Suggestions for Authors

L35-48: An important note on the conclusions (here Results) drawn from the studies cited below: These findings seem highly questionable, as they are based on mouse models with a few exceptions based on human volunteers. By the way, the human studies rather document no radical changes in intestinal microbiota. The results of most studies are often significantly different and incomparable. Please, reword, rephrase, discuss!!

L56 and everywhere below (e.g. L185-162): AS, NNS (nonnutritive sweeteners) is a term for a whole range of sweeteners. Therefore, you use plural verbs?!

L97-98 and everywhere below in the text: please, use the current designation of bacterial kingdoms and phyla based on the LPSN database (https://lpsn.dsmz.de/domain/bacteria, https://pubmed.ncbi.nlm.nih.gov/38252124/).

L169-172: please, reword this unreadable sentence!

Author Response

Comment 1: L35-48: An important note on the conclusions (here Results) drawn from the studies cited below: These findings seem highly questionable, as they are based on mouse models with a few exceptions based on human volunteers. By the way, the human studies rather document no radical changes in intestinal microbiota. The results of most studies are often significantly different and incomparable. Please, reword, rephrase, discuss!!

Response 1: Thank you for this important observation. We have revised the relevant section in the Results.

Comment 2: L56 and everywhere below (e.g. L185-162): AS, NNS (nonnutritive sweeteners) is a term for a whole range of sweeteners. Therefore, you use plural verbs?!

Response2: We appreciate the reviewer’s insightful note. Corrected.

Comment 3:L97-98 and everywhere below in the text: please, use the current designation of bacterial kingdoms and phyla based on the LPSN database (https://lpsn.dsmz.de/domain/bacteria, https://pubmed.ncbi.nlm.nih.gov/38252124/).

Response 3: We thank the reviewer for this helpful suggestion. In response, we have reviewed and updated all bacterial taxonomy throughout the manuscript.

Comment 4:L169-172: please, reword this unreadable sentence!

Response 4: We appreciate the reviewer’s insightful note. Reworded to a more readable sentence.